



# Source and variability of formaldehyde (HCHO) at northern high latitude: an integrated satellite, ground/aircraft, and model study

Tianlang Zhao[1], Jingqiu Mao[1], William R. Simpson[1], Isabelle De Smedt[2], Lei Zhu[3], Thomas F.

Hanisco[4], Glenn M. Wolfe[4], Jason M. St. Clair[4,5], Gonzalo González Abad[6], Caroline R.

Nowlan[6], Barbara Barletta[7], Simone Meinardi[7]  and Donald R. Blake[7]

[1] University of Alaska Fairbanks, Department of Chemistry and Biochemistry & Geophysical Institute, Fairbanks, AK, United States

[2] Royal Belgian Institute for Space Aeronomy (BIRA-IASB), Brussels, Belgium

[3] Southern University of Science and Technology, School of Environmental Science and Engineering, Shenzhen, China

[4] NASA Goddard Space Flight Center, Atmospheric Chemistry and Dynamics Lab, Greenbelt, MD, United States

[5] University of Maryland Baltimore County, Baltimore, MD, United States

[6] Harvard-Smithsonian Center for Astrophysics, Cambridge, MA, United States

[7] University of California Irvine, Irvine, CA, United States





*Correspondence to*: Tianlang Zhao (tzhao@alaska.edu) and Jingqiu Mao (jmao2@alaska.edu)

**Abstract**: Here we use satellite observations of HCHO vertical column densities (VCD) from the

TROPOspheric Monitoring Instrument (TROPOMI), ground-based and aircraft measurements,

combined with a nested regional chemical transport model (GEOS-Chem at 0.5°×0.625° resolution), to

understand the variability and sources of summertime HCHO better in Alaska. We first evaluate GEOS-

Chem with *in-situ* airborne measurements during Atmospheric Tomography Mission 1 (ATom-1)

aircraft campaign and ground-based measurements from Multi-AXis Differential Optical Absorption

Spectroscopy (MAX-DOAS). We show reasonable agreement between observed and modeled HCHO,

isoprene and monoterpenes. In particular, HCHO profiles show spatial homogeneity in Alaska,

suggesting a minor contribution of biogenic emissions to HCHO VCD. We further examine the

TROPOMI HCHO product in Alaska during boreal summer, which is in good agreement with GEOS-

Chem model results. We find that HCHO VCDs are dominated by free-tropospheric background in

wildfire-free regions. During the summer of 2018, the model suggests that the background HCHO

column, resulting from methane oxidation, contributes to 66 to 80% of the HCHO VCD, while wildfires

contribute to 14% and biogenic VOC contributes to 5 to 9% respectively. For the summer of 2019,

which had intense wildfires, the model suggests that wildfires contribute to 40 to 65%, and the

background column accounts for 30 to 50% of HCHO VCD in June and July. In particular, the model

indicates a major contribution of wildfires from direct emissions of HCHO, instead of secondary

production of HCHO from oxidation of larger VOCs. We find that the column contributed by biogenic



VOC is often small and below the TROPOMI detection limit. The source and variability of HCHO

VCD above Alaska during summer is mainly driven by background methane oxidation and wildfires.

This work discusses challenges for quantifying HCHO and its precursors in remote pristine regions.

## 1. Introduction

The Arctic (north of 66.5°N) and boreal region (between 45°N and 65°N) have undergone dramatic

temperature and ecological changes over the past century and the rate of this change has accelerated in

recent decades (Cohen et al., 2014). Satellite-based observations of leaf area index (LAI) and

normalized difference vegetation index (NDVI) suggest that northern high latitudes shows a significant

trend of greening in the past three decades as a result of vegetation growth (Bhatt et al., 2017; Keeling

et al., 1996; Myers-Smith et al., 2011; Myneni et al., 1997; Xu et al., 2013; Zhou et al., 2001; Zhu et al.,

2016), in part because the temperature is the limiting factor for vegetation growth in this region

(Nemani et al., 2003). In the meantime, boreal forest fires have shown an increasing trend over the past

few decades, which is likely to continue (Abatzoglou and Williams, 2016).

Terrestrial vegetation emits a significant amount of volatile organic compounds (VOCs), which play a

major role in air quality and chemistry-climate interactions (Guenther et al., 1995). These biogenic

VOCs (BVOCs) undergo photochemical degradation, leading to the formation of ozone and aerosol

particles that play major roles in climate and air quality (Mao et al., 2018). Biogenic VOCs account for



more than 80% of global VOC emissions and represent a major source of reactive carbon to the

atmosphere (Guenther et al., 1995, 2006). Primary biogenic VOC emissions include both isoprene (2-

methyl-1,3-butadiene, $C_5H_8$) and monoterpenes (a class of terpenes that consist of two isoprene units,

$C_{10}H_{16}$). After these biogenic VOCs are emitted to the atmosphere, HCHO is rapidly produced through

oxidation of isoprene and monoterpenes (Millet et al., 2006; Palmer et al., 2006). The emissions of these

biogenic VOCs are dependent on the air temperature, light intensity, plant functional type (PFT), leaf

area index (LAI), leaf age, soil moisture, ambient carbon dioxide ($CO_2$) concentrations and a number of

other factors (Guenther et al., 2006). It has been suggested that at least some ecosystems in the northern

high latitudes are highly sensitive to temperature, leading to a strong increase in BVOC emissions in

recent years (Kramshøj et al., 2016; Lindwall et al., 2016). BVOC emissions are further complicated by

land cover and LAI changes in this region (Tang et al., 2016).

Biogenic VOC emissions in the Arctic and boreal region are poorly characterized, due to lack of

measurements. Previous measurements have been generally focused on European boreal forests with a

major focus on monoterpenes (Bäck et al., 2012; Juráň et al., 2017; Rantala et al., 2015; Rinne et al.,

2000; Spirig et al., 2004; Zhou et al., 2017). Biogenic VOC emissions in other boreal forests outside of

Europe have seldom been quantified. Some early aircraft-based measurements show abundant isoprene

in Alaskan boreal forests (Blake et al., 1992), suggesting a major gap in current understanding of BVOC

emissions in this region. Isoprene fluxes in tundra systems have been measured in Greenland (Kramshøj



et al., 2016; Lindwall et al., 2016; Vedel-Petersen et al., 2015), northern Sweden (Faubert et al., 2010;

Tang et al., 2016) and the Alaskan North Slope (Angot et al., 2020; Potosnak et al., 2013). All these

tundra measurements show a very strong positive temperature dependence for isoprene fluxes, likely

due to higher emission potentials for isoprenoids than temperate species (Rinnan et al., 2014). This high

sensitivity to temperature suggests an important role of climate warming on BVOC emissions, and

potentially on the air quality and climate.


Formaldehyde (HCHO) serves as an important indicator of BVOC emissions on regional and global

scales (Millet et al., 2006). The HCHO column density has been observed from space by several

satellite sensors including the Global Ozone Monitoring Experiment (GOME) (Palmer et al., 2001),

Scanning Imaging Absorption Spectrometer for Atmospheric Cartography (SCIAMACHY) (De Smedt

et al., 2008), and Ozone Monitoring Instrument (OMI) (González Abad et al., 2015). A number of

studies use satellite-based observations of the HCHO column density to quantify regional and global

isoprene emissions in regions where BVOC emissions are dominated by isoprene (Guenther et al.,

2006; Millet et al., 2008; Palmer et al., 2003, 2006; Stavrakou et al., 2009, 2014), and the interannual

variation of BVOC emissions (De Smedt et al., 2010, 2015; Stavrakou et al., 2014, 2015; Zhu et al.,

2017).  For example, both Bauwens et al.(2016) and Stavrakou et al.(2018) find an increasing trend in

the HCHO column over northern high latitudes, using OMI observations during the period of 2005-

2015.



Biomass burning represents another major source of HCHO from both primary emissions and secondary

production from VOC precursors. Biomass burning is the second largest source of global non-methane

volatile organic compounds (NMVOCs) after biogenic emissions (Yokelson et al., 2008). The GFED4s

burned area dataset including small fires shows that boreal forests are responsible for 2.5% of global

burned area but 9% of fire carbon emission and 15% of fire methane ($CH_4$) emission (van der Werf et

al., 2017). Several studies have reported a high level of HCHO emitted from wildfire plumes. Liu et al

(2017) found formaldehyde as the second most abundant NMVOC from wildfires in western US, with

an emission factor of 2.3 ($\pm$0.3) g/kg dry matter for temperate forests. A similar emission factor was

suggested for boreal forest fires (Liu et al.,2017). As boreal fires have become more intense in the past

few decades, HCHO from boreal fires are likely to play an important role in the temporal and spatial

variability of HCHO in this region.


While satellite-based observations of HCHO appear promising, their application in air quality and

regional photochemical modeling remains challenging. There are large uncertainties and inconsistencies

among different satellite-based sensors and retrieval methods for HCHO, due to instrumental

sensitivity, retrieval algorithms, timing of observation with respect to the diurnal cycle, as well as

several other factors (Smedt et al., 2015; Zhu et al., 2016). Zhu et al (2016) show that differences

among these satellite sensors can be as much as a factor of two, posing a challenge for comparing





different satellite based HCHO observations. Another uncertainty lies in the reference sector correction, which is usually done by subtracting the retrieved SCD measured over the remote Pacific from the retrieved terrestrial SCD observed at the same latitude (Khokhar et al., 2005). The corrected differential

SCD, which we call the dSCD, represents a HCHO enhancement relative to the Pacific background (Zhu et al., 2016). Several studies have shown systematic biases in satellite HCHO products. Wolfe et al. (2019) finds a small bias in OMI HCHO when comparing to ATom-1 and ATom2 datasets. Using FTIR ground-based measurements, Vigouroux et al. (2020) finds a positive bias of 25% in TROPOMI HCHO vertical column density in regions with low HCHO ($<2.5\times10^{15}$ molecules cm$^{-2}$) and a negative

bias of 31% in regions with high HCHO ($>8.0\times10^{15}$ molecules cm$^{-2}$), consistent with a recent comparison between MAX-DOAS and TROPOMI (De Smedt et al., 2021). Zhu et al.(2020) finds a similar bias for OMI HCHO product, with *in-situ* measurements from aircraft campaigns.

Ground-based remote sensing measurements offer a complementary way to measure the column density

of trace gases with several advantages. We here mainly focus on the MAX-DOAS technique, but other techniques are also widely used, such as direct sun measurement (e.g. Pandora instrument) (Cede et al., 2006; Park et al., 2018; Herman et al., 2009), and the ground-based direct solar absorption FTIR (Fourier Transform Infra-Red) technique (Vigouroux et al., 2020, 2009, 2018). First, ground-based measurements provide higher precision than satellite sensors, and ground-based measurements can

increase the signal-to-noise ratio (SNR) by either averaging over a large number of spectra or having a





higher light intensity by looking directly at the sun. With higher precision and accuracy, ground-based measurements can provide constraint for satellite-based observations, reducing the possible errors due to background correction as mentioned above and helping to assess possible biases due to uncertainties from air mass factors and spectral fitting. Second, MAX-DOAS can be made during cloudy days and

low visibility days (Hönninger et al., 2004), while satellite-based measurements are far more difficult to interpret under these conditions. MAX-DOAS uses observations of the slant column of oxygen collisional dimers ($O_2$-$O_2$, also known as $O_4$) to quantify effective path lengths (Frieß et al., 2006), which are then used in the retrieval of HCHO.  These methods, which typically use optimal-estimation inversions, agree well with co-located ceilometer, sun photometer aerosol optical depth (AOD)

measurements and show consistent results with LP-DOAS retrievals (Frieß et al., 2011), although there are complications (Ortega et al., 2016). MAX-DOAS has been extensively used for boundary-layer species such as $NO_2$ (Wagner et al., 2011), HCHO (Heckel et al., 2005; Peters et al., 2012; Vlemmix et al., 2015) and BrO measurements (Simpson et al., 2017, 2018). Recently, De Smedt et al uses a global network of 18 MAX-DOAS instrument to validate large range HCHO columns measured by OMI and

TROPOMI satellite sensors (De Smedt et al., 2021).

Here we use satellite-based observations of HCHO VCDs from TROPOMI, ground-based and aircraft measurements, combined with a high-resolution chemical transport model (GEOS-Chem at $0.5° \times 0.625°$ resolution), to understand the sources and variability of summertime HCHO in Alaska better.





## 2. Observations and Model

### 2.1. TROPOMI

In this study, we use the TROPOMI operational level 2 (L2) HCHO vertical column density (VCD) product, version 1.1.5-7. The TROPOMI sensor, on board the Sentinel-5 Precursor (S5P) satellite, provides a horizontal resolution of 3.5 km × 7 km from 2018 May to 2019 August, 3.5 km × 5.5 km since August 2019. This product provides a continuous record of reprocessed + offline data (RPRO+OFFL) since 2018 May. More details can be found in the S5P TROPOMI HCHO L2 product user manual (Veefkind et al., 2012).

The retrieval algorithm for the S5P TROPOMI HCHO product is based on DOAS technique, following the OMI QA4ECV product retrieval algorithm (http://www.qa4ecv.eu/ecv/hcho-p/data) detailed in De Smedt et al. (2018). The HCHO slant column density ($SCD_{1,SAT}$) is retrieved in the fitting window of 328.5-359 nm (TROPOMI channel 3). The DOAS reference spectrum is based on the spectra averaged over tropical Pacific region from previous day. Therefore, since $SCD_{1,SAT}$ is derived from the difference between local spectra and reference spectrum, it quantifies the slant column exceeding the average Pacific background. The L2 product provides an air mass factor ($AMF_{SAT}$) to convert slant column absorbances of trace gases to vertical column absorbances. $AMF_{SAT}$ is computed from a radiative transport model (RTM) VLIDORT v2.6 (Spurr, 2008) and is dependent on observation geometry,



surface albedo, cloud properties, and the vertical distribution of relevant species. The retrieval uses the

1°×1° monthly averaged surface albedo measured by OMI (Kleipool et al., 2008). A priori vertical

profiles of relevant species are provided by the daily forecast (NRT) or reanalysis of a chemical

transport model, TM5-MP, at $1° \times 1°$ spatial resolution (Williams et al., 2017).

To correct for possible systematic time- and latitude-dependent offsets, a reference sector correction is

applied to calculate the differential slant column, dSCD$_{SAT}$. This correction is based on the assumption

that the background HCHO column over remote oceanic regions is only due to methane oxidation,

which is presumed to be modeled correctly in the TM5-MP CTM. The TROPOMI-measured HCHO

differential slant column, dSCD$_{SAT}$ equals the SCD$_{1, SAT}$ minus the reference sector SCD$_{Ref,SAT}$. The

reference sector SCD$_{Ref,SAT}$ consists of two parts, an across-track correction (the mean SCD$_{1, SAT}$ in the

equatorial reference sector ([-5°,5°], [180°,240°])) and the zonal along-track correction (a polynomial of

all-rows-combined mean SCD$_{1, SAT}$ in 5° latitude bins (only selecting SCD$_{1,SAT}$ that is lower than $5\times10^{16}$

molecules cm$^{-2}$) in the reference sector ([-90°,90°], [180°, 240°])). Technical details can be referred to

De Smedt et al. (2018). The resulting differential column, dSCD$_{SAT}$, is then added to the background

slant column calculated by the TM5-MP CTM, for the tropospheric vertical column (VCD$_{SAT}$):


$$VCD_{SAT} = \frac{dSCD_{SAT}}{AMF_{SAT}} + VCD_{0,SAT} = \frac{SCD_{1,SAT} - SCD_{Ref,SAT}}{AMF_{SAT}} + \frac{AMF_{0,SAT} * VCD_{0,CTM}}{AMF_{SAT}} \qquad (1)$$

Here $SCD_{1,SAT}$ is the measured slant column density, $SCD_{Ref,SAT}$ is the background slant column correction in reference sector. $AMF_{SAT}$ is the air mass factor provided by the TROPOMI HCHO product. $AMF_{0,SAT}$ is the air mass factor for the background column in the reference sector. $VCD_{0,CTM}$ is

the vertical column in reference sector calculated by a CTM model (TM5-MP CTM), in the TROPOMI HCHO product.

Following S5P TROPOMI HCHO L2 user manual (Veefkind et al., 2012), we applied several criteria to ensure the data quality. This includes: (1) quality assurance values (QA) greater than 0.5; (2) cloud

fraction at 340 nm less than 0.5; (3) Solar Zenith Angle (SZA) less than 60°;(4) surface albedo less than 0.1, and (5) derived AMF greater than 0.1. In particular, northern Alaska can be covered by snow and ice even in summer with the criteria of surface albedo. We do not use the data over snow/ice surface as the retrieval algorithm may not work well on these surfaces (De Smedt et al., 2018). We use the overpass data in the local time window 12:00–15:00 AKDT (20:00–23:00 UTC).


To compare the HCHO column density from TROPOMI with our model, we recalculate the AMF based on vertical shapes from GEOS-Chem simulations and scattering weight from TROPOMI HCHO



product. This method has been applied in a number of previous studies (Palmer et al., 2001; Boersma et al., 2004; González Abad et al., 2015; Zhu et al., 2016).


$$AMF_{GC} = \int_{P_s}^{0} \frac{\Omega_{GC}(p)}{\Omega_{A,GC}} w(p)\, dp \qquad (2)$$

Here $\Omega_{GC}(p)$ is the column density of the air parcel at vertical air pressure $p$, for a specific air column. $\Omega_{A,GC}$ is the total column of the specific air column. $w(p)$ is scattering weight of TROPOMI HCHO product at each altitude, calculated by the product of TROPOMI averaging kernel $A_{SAT}(p)$ and air mass factor $AMF_{SAT}$. $P_s$ is surface layer pressure.


The standard TROPOMI HCHO VCD relies on the TM5-MP CTM for background HCHO fields, but in this work, we are using a different CTM, GEOS-Chem. Therefore, to be consistent with the GEOS-Chem CTM, we reprocessed TROPOMI HCHO vertical column that by replacing the original background ($VCD_{0,SAT}$ in Equation (1)) with $VCD_{0,GC}$ from GEOS-Chem background simulation

(González Abad et al., 2015; Kaiser et al., 2018).We assume that background is approximately equal to GEOS-Chem $VCD_{0,GC}$ and neglect the variability of the $AMF_{0,GC}/AMF_{GC}$ ($AMF_{0,GC}$ is $AMF_{GC}$ in reference sector averaged in 5° latitude bins) (De Smedt et al., 2018). The GEOS-Chem background simulation is performed over reference sector and excludes biogenic and biomass burning emissions (Table 1). Finally, the reprocessed TROPOMI HCHO VCD is expressed as:






$$VCD_{SAT,GC} = \frac{dSCD_{SAT}}{AMF_{GC}} + VCD_{0,GC} \quad (3)$$

Our reprocessed TROPOMI HCHO VCD might have several advantages over the TROPOMI HCHO

operational product that is based on TM5-MP model. First, our reprocessed HCHO VCD are based on

GEOS-Chem nested simulation with finer resolution than TM5-MP model. Second, our GEOS-Chem

simulation includes year-specific wildfire emissions that was not available for TM5-MP model when

TROPOMI operational product was produced. As we show below, this reprocessed HCHO VCD shows

higher values in central Alaska than the original product, leading to a better agreement with model

results.

### 2.2. ATom-1 aircraft campaign

The NASA Atmospheric Tomography (ATom) studied atmospheric composition in remote regions

(Wofsy et al., 2018). ATom had four phases over a 4-year period, with each phase sampling the global

atmosphere in one of four seasons. ATom deployed a comprehensive gas and aerosol particle

measurement payload on the NASA DC-8 aircraft. During ATom-1, two flights performed vertical

profiling over Alaska during August 1–3 in 2016. We make use of HCHO measurement by Laser

Induced Fluorescence technique (Cazorla et al., 2015) and VOC measurement by whole air sampling

(WAS) followed by laboratory Gas Chromatography (GC) analysis (Simpson et al., 2020) during these

flights to evaluate model performance on HCHO, isoprene and monoterpenes ($\alpha$-pinene and $\beta$-pinene).



We use 1 minutes averaged data for HCHO and 3-5 minutes average data for isoprene and

monoterpenes. The reported measurement uncertainties are 10% for HCHO and 10% for isoprene and

monoterpenes.

### 2.3. MAX-DOAS

The Multi-AXis Differential Optical Absorption Spectroscopy (MAX-DOAS) measurement technique

is employed to measure atmospheric trace gases such as HCHO at urban and remote sites (Honninger,

2004). Previous HCHO measurements by MAX-DOAS spectroscopy provided ground validation for

satellite HCHO retrievals and model results (Pinardi et al., 2013). A comprehensive description of

MAX-DOAS retrieval algorithm theoretical basis can be found in Honninger et al (2004).

In this study, we use HCHO VCD timeseries from two MAX-DOAS instruments deployed in Fairbanks

(roof of Geophysical Institute on University of Alaska Fairbanks campus, 64.84° N, 147.72° W) and

Toolik Field Station (TFS, 68.626°N, 149.603°W) since 2017 for Fairbanks and 2019 for TFS.

Fairbanks is in the central Alaska boreal forest region, and TFS is located on the North Slope and is

covered by tundra. The two instruments sample profiles every 12 minutes, which are averaged to 2-hour

intervals for this work and are selected to be in the 12:00-15:00 local time window. In this study, the

VCD is calculated using the formula $VCD = dSCD_{20°} / dAMF_{20°}$ where the $dSCD_{20°}$ is the measured

differential slant columns, the difference in HCHO absorption between a 20° elevation angle and the

zenith view. The $dAMF_{20°} = 1.93$ is calculated geometrically (Ma et al., 2013), and 20° was chosen as



the highest elevation in all measurement sequences. Wildfire polluted records are removed by selecting

UV visibility > 5 km; foggy records are removed by selecting dewpoint depression < 1.5 °C (at

Fairbanks) or < 80% RH (at TFS). The footprint of MAX-DOAS is about 20 km for clear-sky

conditions, shorter with clouds or high particulate loading. The 2σ detection limit of VCD = $1.0 \times 10^{15}$

molecules $cm^{-2}$ by this geometric method.

### 2.4. Nested GEOS-Chem simulation

Here we use GEOS-Chem v12.5.0 (doi: 10.5281/zenodo.3403111). GEOS-Chem is a 3-D global

chemical transport model driven by Modern-Era Retrospective analysis for Research and Applications,

Version 2 (MERRA-2) by the Global Modeling and Assimilation Office (GMAO) at NASA's Goddard

Space Flight Center (Rienecker et al., 2011), at a horizontal resolution of 0.5° × 0.625° and 72 vertical

layers from surface to 0.01 hPa. GEOS-Chem v12.5.0 provides a new nested capability, FlexGrid,

allowing users to define the model grid at run time (http://wiki.seas.harvard.edu/geos-

chem/index.php/FlexGrid). We take advantage of this nested capability to investigate the spatial

variability of HCHO and VOCs over Alaska domain (170°W–130°W, 50°N–75°N), at a horizontal

resolution of 0.5° × 0.625°. The boundary conditions for the nested run are updated every 6 hours, from

GEOS-Chem global simulation at 2° × 2.5° with the same model configuration. The nested simulation

was conducted for two summers (May 1 to August 31) in 2018 and 2019.


Biomass burning emissions follow the preliminary 'beta' version of Global Fire Emission Database,

GFED4.1s biomass burning emissions processed for GEOS-Chem (Giglio et al., 2013). We use monthly

average emissions calculated in GFED4.1s based on fire detection and burning area from MODIS

satellite (van der Werf et al., 2017). The biomass burning emissions in 2018 and 2019 has been updated

to reflect the year-specific emissions.

BVOC emission in the model follows the Model of Emissions of Gases and Aerosols from Nature

(MEGAN, v2.1) (Guenther et al., 2006, 2012). In this work, BVOC emission activity factors are

calculated online, expressed as:

$$\gamma = C_{ce} \cdot LAI \cdot \gamma_P \cdot \gamma_T \cdot \gamma_A \cdot \gamma_{SM} \cdot \gamma_{CO_2}$$

Here $C_{ce}$ is a standard environment coefficient normalizing $\gamma$ to 1 under standard environmental

condition. LAI is the leaf area index ($m^2\,m^{-2}$), $\gamma_P$ and $\gamma_T$ are emission activity factors accounting for

light and temperature effects, respectively. $\gamma_P$ is calculated based on the photosynthetic photon flux

density (PPFD) (µmol of photons in 400–700 nm range $m^{-2}\,s^{-1}$). Terrestrial vegetation for BVOC

emissions is based on the plant functional type (PFT) distribution derived from Community Land Model

(CLM4) (Lawrence et al., 2011; Oleson et al., 2013). CLM4 output suggests two dominating PFTs in

the continent of Alaska: needle leaf evergreen boreal tree (mainly in the interior boreal forest region)





and broadleaf deciduous boreal shrub (mainly over north slope and southwest Alaska), both with high

emission factors in isoprene (3000 µg m$^{-2}$ h$^{-1}$ and 4000 µg m$^{-2}$ h$^{-1}$ respectively) and low EFs in

monoterpenes ($\alpha$-pinine + $\beta$-pinine, 800 µg m$^{-2}$ h$^{-1}$ and 300 µg m$^{-2}$ h$^{-1}$ respectively). Thus, we expect a

major contribution from isoprene to BVOC emissions in Alaska in model results. Despite that shrub has

a higher emission factor of isoprene, we expect a larger isoprene emission flux from central Alaska

boreal forest region due to warmer "continental" temperatures and higher LAI.

In this work we use the detailed $O_3$-$NO_x$-$HO_x$-VOC chemistry ("tropchem" mechanism) (Park et al.,

2004; Mao et al., 2010, 2013), with updates on isoprene chemistry (Fisher et al., 2016). This version of

isoprene chemistry in GEOS-Chem have been extensively evaluated by recent field campaign data and

satellite observations (Fisher et al., 2016; Travis et al., 2016), including HCHO production from

isoprene oxidation (Zhu et al., 2016, 2020, Kaiser et al. 2018). In general, under high-$NO_x$ condition (1

ppbv), the HCHO production is prompt, reaching 70-80% of its maximum yield within a few hours.

While under low-NOx condition (0.1 ppbv or lower), it takes several days to reach the maximum yield

and the cumulative yield is still lower than the high-NOx condition by a factor of 2–3 (Marais et al.,

2012). As we show below, this slow production of HCHO under low-$NO_x$ conditions leads to weak but

widespread HCHO enhancement in regional scale.






To examine the influence of different sources on HCHO columns in Alaska, we conducted a series of

nested GEOS-Chem simulations, as described in Table 1. The background HCHO column ($VCD_{0,GC}$) is

calculated from a GEOS-Chem simulation where biogenic emission and biomass burning emission are

turned off. The HCHO differential column induced by wildfire or biogenic emission is derived from the

difference between the control run and the run with wildfire or biogenic emission turned off.

**Table 1| Configurations of GEOS-Chem simulations in this study.**

| Simulations | Biogenic emission | Wildfire |
|---|---|---|
| Control (Ctrl) | On | On |
| Background (BG) | Off | Off |
| No Fire (NF) | On | Off |
| No biogenic emission (NB) | Off | On |

## 3.  Model evaluation by ATom-1 and surface measurements

### 3.1. Model evaluation by ATom-1

Figure 1 (a – c) shows measured vertical profiles of formaldehyde, isoprene and monoterpenes across

the Alaska domain during ATom-1. In Figure 1(a), the measured HCHO mixing ratio decreases





exponentially from surface (around 320 pptv) to the upper troposphere (around 100 pptv). The HCHO

surface mixing ratio in Alaska is an order of magnitude lower than other high-BVOC regions such as

Southeast US (Li et al., 2016).

Figures 1(b) and (c) show that observed isoprene and monoterpenes have much higher mixing ratios in

the lowest 2 km layer than above. The mean observed isoprene mixing ratio is about 120 pptv in the

boundary layer, a factor of three higher than that of monoterpenes. As isoprene has a shorter lifetime

(1.1 hours) than monoterpene (2.1 hours), this indicates a stronger isoprene emission flux than

monoterpene emission flux in Alaskan boreal forest. The predominance of isoprene emission in Alaskan

boreal forest is different from some European boreal forests, where monoterpenes are often the

predominant BVOC species (Juráň et al., 2017; Bäck et al., 2012).


To evaluate model performance with ATom-1 measurements, a nested GEOS-Chem simulation is

conducted during ATom-1 mission period over Alaska. We sampled the model along the flight track at

the flight time with 1-hour model time resolution for comparisons between model and observations. As

shown in Figure 1(d) – (f), our nested GEOS-Chem model well reproduce the ATom-1 vertical and

spatial variability of HCHO, isoprene and monoterpenes mixing ratios. Modeled isoprene and

monoterpenes mixing ratios concentrate in the surface layer (0 – 2 km) and show a median value of

around 100 pptv and 10 pptv respectively. Modeled isoprene mixing ratio is comparable with ATom-1





observations, while monoterpenes mixing ratio is lower than ATom-1 averaged value in lower than 2 km (around 40 pptv).


One remarkable feature in Figure 1 is the spatial homogeneity in HCHO vertical profiles, as shown in both observations and model. We find that all sampled HCHO vertical profiles in Alaska show similar magnitude and vertical distribution, despite different land types and locations of these sampled profiles. The homogeneity is not observed in isoprene and monoterpene mixing ratios, which show maximums in

central and south Alaska, where boreal forests are located (Figure S1). Such spatial discrepancies between HCHO and isoprene/monoterpenes suggest a minor contribution of biogenic VOC emissions to HCHO column density.

We further examine the abundance of isoprene and monoterpenes in Alaska with available surface VOC

measurements from field campaigns at TFS. Angot et al. (2020) reported surface-level ambient mixing ratios of isoprene (0–505 pptv, mean of 36.1 pptv) and monoterpenes (3–537 pptv, 14±18 pptv; median ± standard deviation) in 2018 and 2019 summer. GEOS-Chem is in reasonable agreement with measurement at TFS, with mean isoprene and monoterpene mixing ratios of 151 pptv and 7 pptv respectively, during corresponding measurement periods. Both field measurements and model suggest

that isoprene is the predominant BVOC in this region.

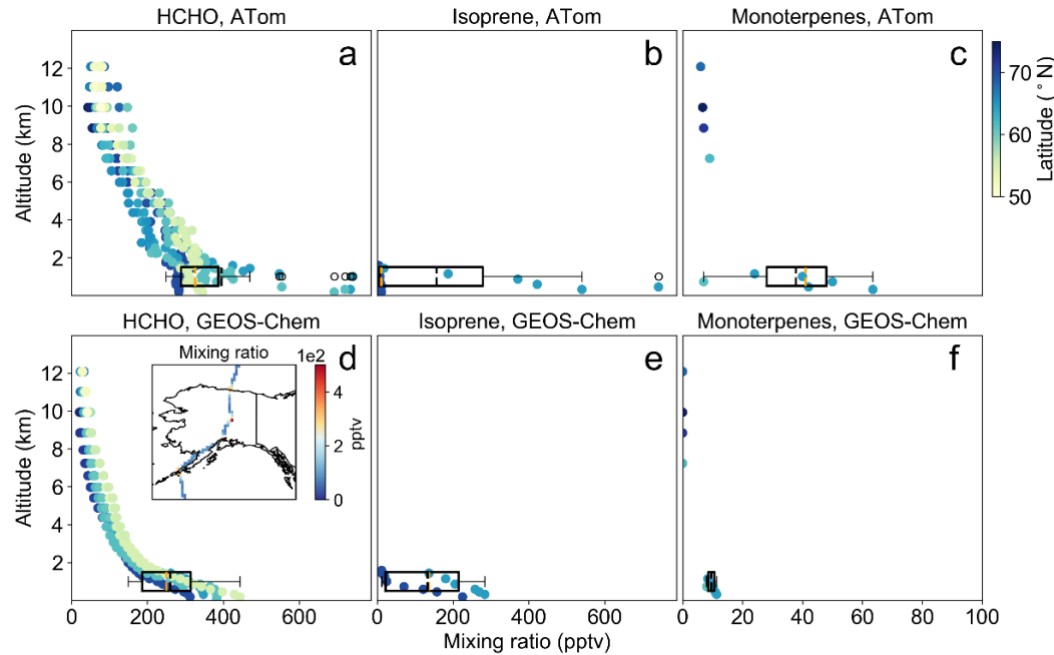

*Figure 1* | *Vertical profiles of HCHO, isoprene and monoterpenes mixing ratio from ATom-1 and*
*GEOS-Chem, along ATom-1 track from 2016 August 1ˢᵗ 19:00 to August 2ⁿᵈ 01:00 UTC. GEOS-Chem*
*data is resampled along ATom-1 track, and ATom-1 data is regridded in GEOS-Chem resolution. (a) to*
*(c) are from GEOS-Chem simulation, (d) to (f) are from ATom-1. The subpanel in (d) shows GEOS-*
*Chem HCHO mixing ratio along ATom-1 track crossing Alaska. Box plots represent data distribution*
*lower than 2km. Orange dashes show the median values, black dashes show the mean values.*



**3.2. Model evaluation by MAX-DOAS**

The MAX-DOAS technique is most sensitive to boundary layer species and loses sensitivity

significantly above the first kilometer or two. The ATom vertical profiles shown in Figure 1 indicate

that "background" HCHO has a large fraction of its column well above the lowest kilometers, which

would be detected with a lower sensitivity. This effect appears to make the simple geometric

approximation used here to retrieve HCHO have a sensitivity more like the differential VCD calculated

in GEOS-Chem (dVCD$_{GC}$), which also mainly concentrates in the < 2 km layer (Figure S4). For this

reason, here we compare MAX-DOAS HCHO total column retrieval VCD$_{MD}$ with GEOS-Chem

dVCD$_{GC}$. A detailed comparison between GEOS-Chem and MAX-DOAS retrievals using an optimal

estimation method that appropriately deals with the reduced sensitivity aloft will be described in a

follow-up study.

Figure 2 shows hourly time series of HCHO VCD from MAX-DOAS (VCD$_{MD}$) and GEOS-Chem

dVCD$_{GC}$ at Fairbanks and TFS in two summers (2018 and 2019). In summer of 2018 (Figure 2(a)),

monthly $VCD_{MD}$ in Fairbanks is below the $1.0\times10^{15}$ molecules cm$^{-2}$ detection limit in May and August

and is $1.0–2.0\times10^{15}$ molecules cm$^{-2}$ in June and July, with showing enhanced VCD$_{MD}$ in summer

months. The summertime increases in GEOS-Chem dVCD$_{GC}$ agree well with the seasonal trend of

VCD$_{MD}$. The GEOS-Chem simulations show large enhancements during wildfire periods, which are

also observed in raw MAX-DOAS data. However, the retrieval of the VCD by MAX-DOAS using the




geometric method requires good visibility so as to see though the tropospheric column and the MAX-

DOAS visibilty data cut elimintates $VCD_{MD}$ data during most wildfire events, leading to the appearance

that MAX-DOAS observes less wildfire-related HCHO. Further analysis of the MAX-DOAS data will

allow better comparison for wildfire-influenced HCHO.

Figure 2 also shows good correlation between $VCD_{MD}$ and modeled isoprene emission $E_{ISOP}$ in

Fairbanks for both 2018 (R = 0.59) and 2019 (R = 0.63). As $VCD_{MD}$ shows a much stronger

temperature dependence than GEOS-Chem background $VCD_{0,GC}$ (Figure S2), $VCD_{MD}$ in Fairbanks

appears to be mainly driven by biogenic emissions. $VCD_{MD}$ and $E_{ISOP}$ is less correlated at TFS (R =

0.43), likely due to weaker emission and lower HCHO VCD.

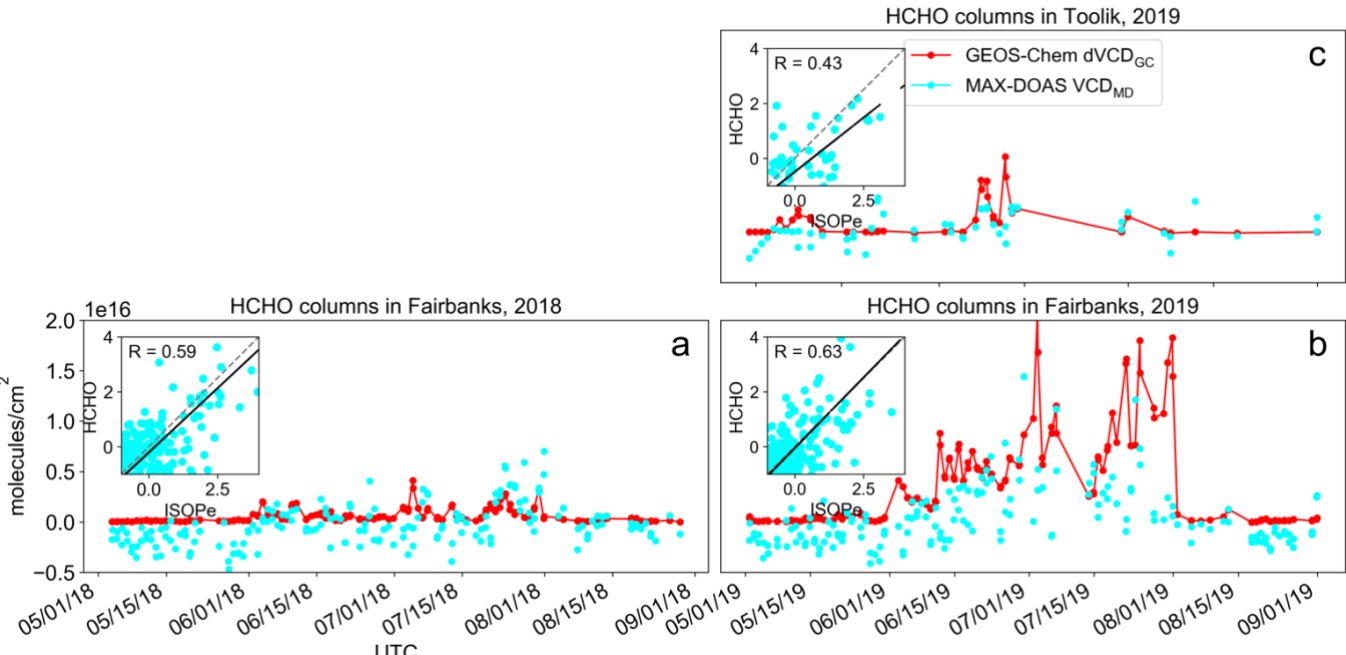

**Figure 2 |** *HCHO dVCD time series in Fairbanks (a and b) and Toolik Field Station (c) in 2018 (mild*

*wildfire) and 2019 summer (severe wildfire), GEOS-Chem simulation (red dots) versus MAX-DOAS*

*measurements (cyan dots). Modelled Fairbanks time series for 2019 are a regional average from a*

*100km×100km domain centering at (64.84° N, 147.72° W), timeseries in Toolik Field Station is similar*

*but centering at (68.626°N, 149.603°W). Subpanels in each panel show the standard major axis (SMA)*

*regression between standardized (z-score) GEOS-Chem isoprene emission and MAX-DOAS HCHO*

*VCD timeseries in each site, as well as the corresponding linear correlation coefficient.*

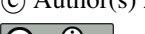



## 4. Evaluating TROPOMI HCHO product

In this section we evaluate the TROPOMI HCHO product over Alaska during the summer of 2018 and

2019. As noted above, these two years differ substantially on local wildfire emissions, providing useful

information on satellite capability of detecting biogenic and wildfire HCHO in remote regions.

### 4.1. Predominance of background chemistry in mild wildfire summer

In Figure 3(a), we show reprocessed monthly TROPOMI HCHO vertical column density ($VCD_{SAT,GC}$)

in Alaska during May-August of 2018 (reprocessing method see section 2.1). Several regions show high

HCHO VCD levels, including central Alaska boreal forest region (Figure S1), with $VCD_{SAT,GC}$ as

$4.6\times10^{15}$ molecules cm$^{-2}$ in July; and north slope and Gulf of Alaska, with $VCD_{SAT,GC}$ as $3\times10^{15}$

molecules cm$^{-2}$ in July.

To understand the drivers for HCHO variability, we first examine the background HCHO VCD

provided by GEOS-Chem ($VCD_{0,GC}$). Figure 3(b) shows that from 2018 May to August, $VCD_{0,GC}$ in

central Alaska increases from $2.0\times10^{15}$ molecules cm$^{-2}$ to $3.5\times10^{15}$ molecules cm$^{-2}$, then decreases to

$2.6\times10^{15}$ molecules cm$^{-2}$, accounting for 66%–80% of $VCD_{SAT,GC}$. This indicates that $VCD_{SAT,GC}$ is

largely dominated by background signals $VCD_{0,GC}$ in 2018. The spatial pattern of $VCD_{0,GC}$, most

noticeable in July, is largely driven by the geography in Alaska. As the majority of HCHO VCD stems





from lowest atmospheric layers (Figure 1), the high elevation in the Alaska Range in southern Alaska

(63°N, 151°W, peaks at Denali, elevation 6190 m) and the Brooks Range in northern Alaska (68°N,

152°W, peaks at Mount Isto, elevation 2736 m) are responsible for the significantly lower HCHO VCD

in these regions. We also find high $VCD_{0,GC}$ (2.0–3.2×10$^{15}$ molecules cm$^{-2}$) over northern Pacific in July

and August, due to enhanced methane oxidation via $CH_3O_2$ + NO reactions near surface and $CH_3O_2$ +

$CH_3O_2$ at higher altitudes. This enhanced methane oxidation also leads to temperature dependence of

$VCD_{0,GC}$ (Figure S2).

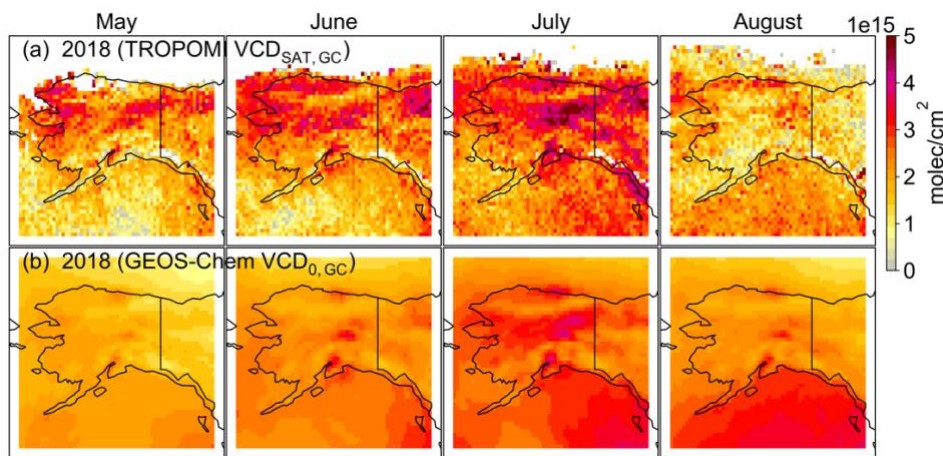

*Figure 3| Reprocessed TROPOMI HCHO VCD and background in 2018 summer. (a) reprocessed*

*TROPOMI HCHO VCD, (b) HCHO background $VCD_0$ used in reprocessed TROPOMI product,*

*provided by GEOS-Chem. GEOS-Chem results are applied the same local noon time window (12:00-*

*15:00) AKDT. TROPOMI data is regridded to GEOS-Chem output spatial resolution (0.5°×0.625°).*





### 4.2. Evaluating TROPOMI HCHO dVCD

Now we further examine measured HCHO signals other than modeled background. Figure 4(a) shows a
monthly spatial pattern of TROPOMI differential HCHO vertical column (dVCD$_{SAT,GC}$ = VCD$_{SAT,GC}$ −
VCD$_{0,GC}$), persistent throughout the 2018 summer. In 2018 July, monthly dVCD$_{SAT,GC}$ is positive over
central Alaska ($1.0 \times 10^{15}$ molecules cm$^{-2}$) and north slope ($4.1 \times 10^{14}$ molecules cm$^{-2}$) and is negative
over southwest Alaska and Gulf of Alaska ($-4.7 \times 10^{14}$ molecules cm$^{-2}$). This pattern is also seen in 2019
summer outside wildfire region.

To quantify the sources of HCHO dVCD, we derive two variables: dVCD induced by wildfire emission
(dVCD$_{GC,Fire}$) and biogenic emission (dVCD$_{GC,Bio}$), computed by the differences between model control
run and sensitivity runs with wildfire or biogenic emissions turned off (Table 1).

We show in Figure 4(c) that dVCD$_{GC,Bio}$ presents a similar spatial pattern and monthly cycle as modeled
isoprene emission (Figure S6), with high values over central boreal forest region ($4.6 \times 10^{14}$ molecules
cm$^{-2}$) and low values in other parts ($5.0–8.0 \times 10^{13}$ molecules cm$^{-2}$). The widespread biogenic HCHO
enhancement can be in part explained by the slow photooxidation in Alaska under low NO$_x$ conditions
(25~35 pptv near surface in GEOS-Chem). Indeed, the HCHO production from isoprene and
monoterpene emissions are lower under low NO$_x$ conditions than high NO$_x$ conditions by a factor of 10



after 24-h oxidation, and it only reaches 20% of its 5-day cumulative yield (Marais et al., 2012). As a result, $dVCD_{GC,Bio}$ in Alaska is lower than that in mid-latitude by more than a factor of 10 for the same amount of isoprene emissions.

Despite the relatively weak Alaskan fire in 2018 summer, we find a higher fraction of $dVCD_{GC,Fire}$ than $dVCD_{GC,Bio}$ in total $dVCD_{GC}$. Figure 4(b) shows several regions with high $dVCD_{GC,Fire}$ ($1.0 \times 10^{15}$ molecules cm$^{-2}$), often co-located with fire hot spots. The GFED4s burning area measured by MODIS is shown in Figure S5. A model sensitivity test in 2018 suggests that over 90% of $dVCD_{GC,Fire}$ is from wildfire direct emission, instead of secondary production of HCHO from oxidation of other VOCs. It is

partly due to the missing of wildfire VOC emissions (Akagi et al., 2011) and the underestimation of secondary wildfire VOC oxidation (Liao et al., 2021; Alvarado et al., 2020). The predominance of combustion HCHO in $dVCD_{GC,Fire}$ is consistent with the strong localization of $dVCD_{GC,Fire}$ enhancement, as the HCHO lifetime is on the order of hours in the presence of sunlight. This also explains why weak wildfire emission (46 GgC) can leads to a stronger HCHO dVCD than biogenic

emission (268 GgC) does.

During 2018 summer, $dVCD_{GC,Fire}$ contributes to 14–22% of $VCD_{GC}$, while $dVCD_{GC,Bio}$ contributes to 6–9% of $VCD_{GC}$ despite that biogenic carbon emissions are higher than wildfire emissions by a factor



of 6. Wildfire and biogenic emission are both important for dVCD$_{GC}$ and most active in central boreal

forest region, posing a challenge to attribute TROPOMI dVCD$_{SAT,GC}$ to individual sources.

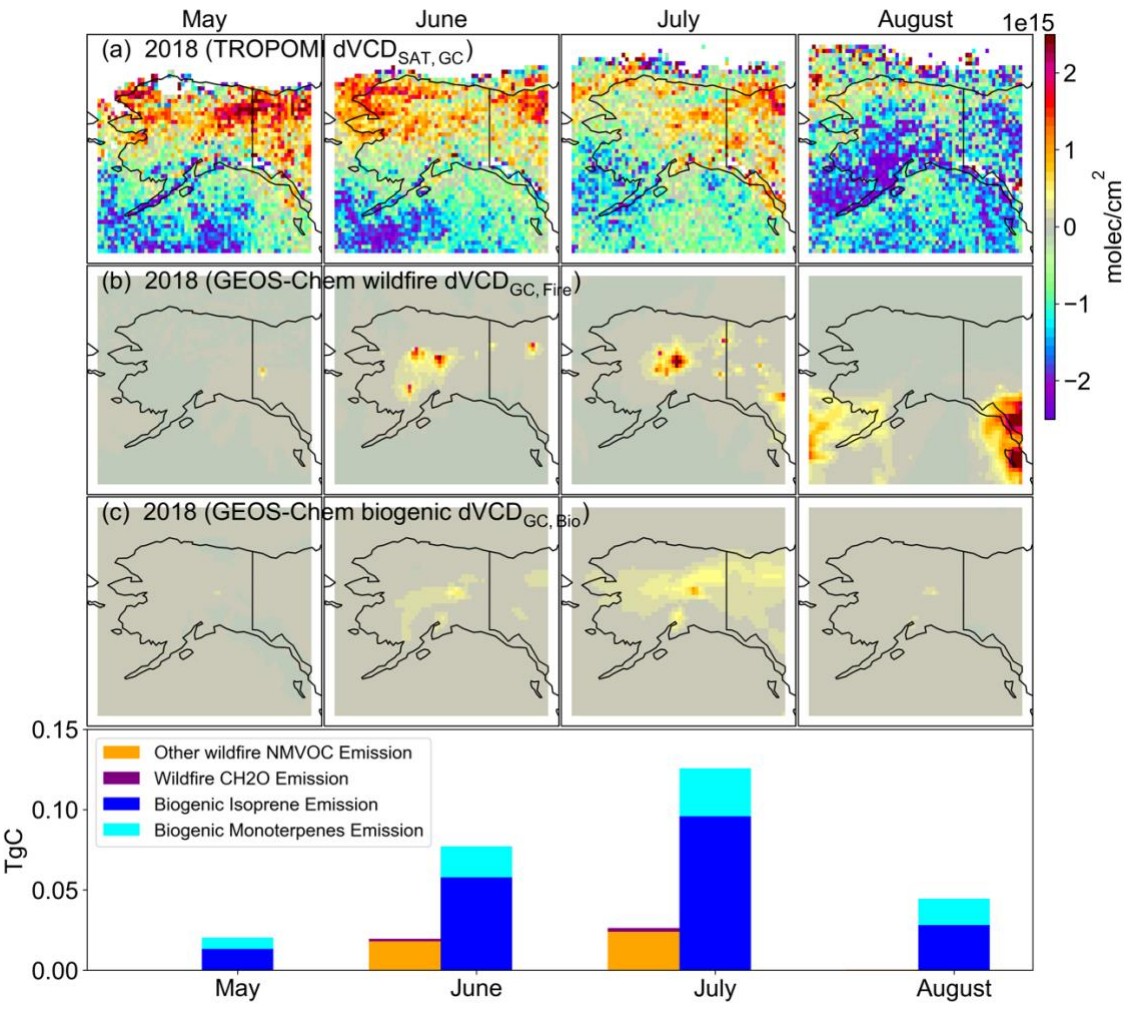



***Figure 4| dVCD and emission in Alaska in 2018 summer.** The first-row panels are TROPOMI monthly*

*HCHO dVCDs in May, June, July and August (unit: molecules cm$^{-2}$), The second row are GEOS-Chem*

*wildfire emission induced monthly $dVCD_{Fire}$. The third-row panels are GEOS-Chem biogenic emission*

*induced monthly dVCD$_{Bio}$. The fourth row is total NMVOC carbon emission from terrestrial vegetation*

*and biomass burning in Alaska in each month (unit: TgC). In (d), blue bars are biogenic isoprene*

*emission, cyan bars are biogenic monoterpenes emission, purple bars represent wildfire HCHO*

*emission, orange bars represent other NMVOCs emitted by wildfire.*

### 4.3. Wildfire emission impacts HCHO column in Alaska

We further examine the summer of 2019. Figure 5(a) shows monthly VCD$_{SAT,GC}$ in 2019 Alaska

summer. In contrast to 2018, TROPOMI observations show an extensive HCHO VCD enhancement

over central Alaska in 2019 July. The monthly average value reaches $1.0×10^{16}$ molecules cm$^{-2}$,

significantly higher than TROPOMI HCHO detection limit (individual scene around $5.0×10^{15}$

molecules cm$^{-2}$). In Figure 5, GEOS-Chem VCD$_{GC}$ reproduces the spatial and temporal variation of

VCD$_{SAT,GC}$ for the summer of 2019. VCD$_{GC}$ shows a monthly HCHO VCD value of $1.2×10^{16}$ molecules

cm$^{-2}$ in central Alaska for July of 2019, similar to VCD$_{SAT,GC.}$ The spatial pattern of VCD$_{GC}$

enhancements agree well with burned area (Figure S5(b)), indicating that the enhancements of



VCD$_{SAT,GC}$ and VCD$_{GC}$ in 2019 July are both strongly induced by wildfire source. Much lower HCHO

VCD are found outside central Alaska and in other months, where there is little fire activity.

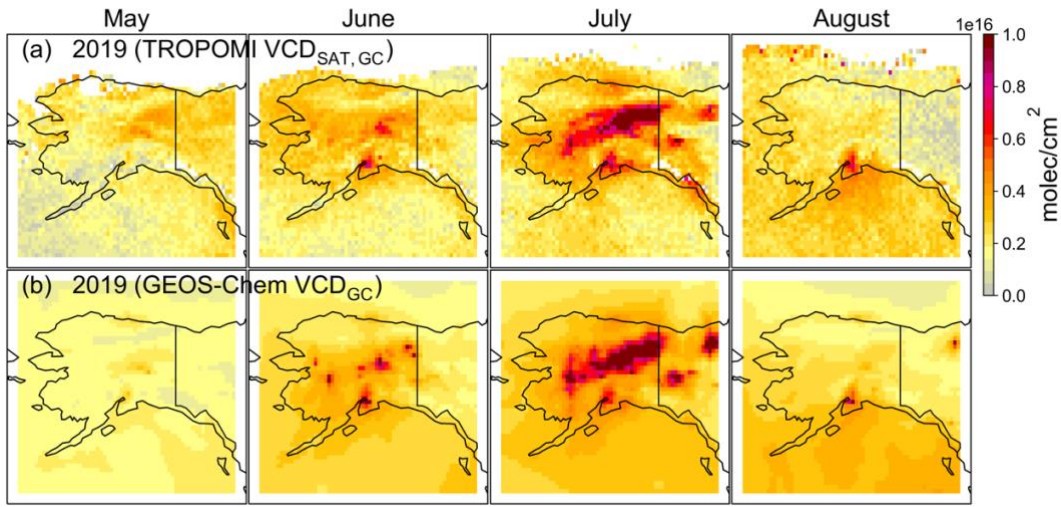


***Figure 5| Reprocessed TROPOMI HCHO VCD and GEOS-Chem HCHO VCD in 2019 summer.***

*TROPOMI HCHO product after August 5, 2019 is upgraded to 5.5km×3.5km resolution.*

Further detailed examination shows the predominance of wildfire emissions on central Alaskan HCHO

VCD in 2019 summer. We find little change on VCD$_{0,GC}$ and dVCD$_{GC,Bio}$ from 2018 to 2019 summer in

model sensitivity tests, while dVCD$_{GC,Fire}$ appears to be solely responsible for the 2018-2019 HCHO

VCD difference especially in July. As a result, dVCD$_{GC,Fire}$ accounts for 65% of VCD$_{GC}$ in central

Alaska, while dVCD$_{GC,Bio}$ only takes 5% of VCD$_{GC}$ and background oxidation accounts for 38% of





$VCD_{GC}$. We emphasize that modeled direct emissions of HCHO from wildfires contributes to 56% of

dVCD$_{GC,Fire}$. Consequently, dVCD$_{GC,Fire}$ is higher than dVCD$_{GC,Bio}$ by a factor of 10 in 2019 Alaska

summer, despite that NMVOC from wildfires (498 GgC) are only higher than biogenic emissions (374

GgC) by a factor of 1 to 2.





*Figure 6| HCHO dVCD and emission in Alaska in 2019 summer.* *Similar to Figure 4 but in 2019. The*

*y-axis range of carbon emission panel is larger than that in Figure 4.*



### 4.4. Uncertainty and capability of TROPOMI in capturing biogenic emission HCHO signals

The total uncertainty of TROPOMI HCHO $VCD_{SAT}$ in Alaska is composed of random and systematic

uncertainties and contributed by errors in dSCD_SAT, AMF_SAT and VCD_{0,SAT}. According to TROPOMI

HCHO ATBD, systematic uncertainties from AMF_SAT accounts for 30–50% of total columns, while the

total contribution of AMF_SAT uncertainties is around 75% of total column uncertainty. We expect the

uncertainty in AMF_SAT to be larger for pixels containing fire smoke due to errors in a priori profile (Zhu

et al., 2020). We find that for regions with heavy smoke, our calculated GEOS-Chem AMF_GC is 50%

lower than TROPOMI AMF_SAT provided, due to the difference in HCHO a priori profiles (Figure S3),

suggesting large uncertainty in the HCHO a priori vertical profiles used for retrieval. In 2019 July,

VCD_{SAT,GC} in central Alaska is enhanced by $3.0–5.0\times10^{15}$ molecules cm$^{-2}$ than VCD_SAT after applying

the AMF_GC that based on GEOS-Chem HCHO profile a priori (Figure S10). Scattering and absorbing

aerosols can also introduce large uncertainties to HCHO AMF by changing the observed radiance

(Gonzi et al., 2011; Jung et al., 2019), especially over strong biomass burning scenes when AMF can be

very sensitive to the vertical profiles of aerosols (Barkley et al., 2012; Fu et al., 2007). $AMF_{SAT}$ errors

can also be due to errors of radiation transfer model and other external parameters like cloud fraction,

surface albedo etc. The aerosol and cloud-related error can be as high as 30% of total columns, due to

the relative low aerosol layer height (around 1 km) of the wildfire smoke in Alaska (Jung et al., 2019).





$dSCD_{SAT}$ contributes to the second most majority of random and systematic errors in VCD$_{SAT}$. The

systematic error in dSCD$_{SAT}$ contributed by reference sector correction leads to the bias pattern of

dSCD$_{SAT}$ in Figure 4(a) and Figure 6(a), within the error range of reference sector correction (0–

$4.0{\times}10^{15}$ molecules cm$^{-2}$) in TROPOMI HCHO Algorithm Theoretical Basis Document (ATBD). The

negative $dVCD_{SAT}$ over southwest Alaska and Gulf of Alaska can be partly contributed by

overcorrection in processing dSCD$_{SAT}$. During the correction, $dSCD_{SAT}$ values higher than $5.0{\times}10^{16}$

molecules cm$^{-2}$ are removed to remove wildfire signals, but the criteria is much higher than wildfire

related HCHO $dVCD_{SAT}$ enhancement in Alaska. According to the bias found in $dVCD_{SAT}$, The

systematic slant columns uncertainty in Alaska can contribute higher than 25% of $VCD_{SAT}$.

Based on ATBD, the uncertainty in VCD$_{0,SAT}$ is estimated to be $0.5–1.5{\times}10^{15}$ molecules cm$^{-2}$ for

difference in HCHO background among different models, accounting for around 40% of $VCD_{SAT}$ in

Alaska summer.


Overall, in Alaska summer, assuming that GEOS-Chem HCHO AMF$_{GC}$ and VCD$_{0,GC}$ share similar

uncertainty with AMF$_{SAT}$ and VCD$_{0,SAT}$ in TROPOMI HCHO operational product, we estimate the total

uncertainty of reprocessed TROPOMI HCHO vertical column to be ≥ 90% for fire free region and ≥

35% for strong wildfire influenced region. The uncertainty range agrees with previous studies about the



biases in satellite HCHO products. Vigouroux et al. (2020) found that TROPOMI HCHO product is

overestimated (26% ±5%) under low HCHO levels and is underestimated (-30.8% ±1.4%) under high

HCHO levels. The bias remains largely uncertainty at high latitude sites, such as in Eureka, Thule, Ny-

Alesund and Lauder. Zhu et al. (2020) found negative biases (-44.5% to -21.7%) under elevated HCHO

columns and positive biases (66.1% to 112.1%) under low HCHO columns in OMI HCHO product,

which is close to the uncertainty estimated for TROPOMI HCHO product. Wolfe et al. (2019) also

found a bias in OMI HCHO when comparing to the whole ATom 1-2 dataset. Since Alaska lies in the

reference sector defined by most retrieval algorithms (González Abad et al., 2015; De Smedt et al.,

2018), errors in background correction can lead to bias in corrected slant column $dSCD_{SAT}$, not only in

Alaska but also in all northern high latitudes region.


## 5. Conclusions and discussions

The Arctic / boreal terrestrial ecosystem is undergoing rapid changes in recent decades, but VOC

emissions from Arctic and boreal vegetation and wildfires remains poorly quantified, limiting our

capability for understanding biosphere-atmosphere exchange in this region and its feedback on Arctic

climate and air quality. HCHO serves as an important indicator for biogenic and wildfire VOC

emissions. In this work, we use satellite-based observations of HCHO VCD from the TROPOMI

instrument on-board S5P satellite, and ground-based measurements of HCHO VCD from MAX-DOAS,





combined with a nested grid regional chemical transport model (GEOS-Chem at (0.5°×0.625°), to

examine source and variability of HCHO VCD in Alaska for the summers of 2018 and 2019.


We first evaluate the GEOS-Chem nested simulation with *in-situ* airborne measurements from the

August 2016 ATom-1 mission and ground-based MAX-DOAS observations in summer 2018 and 2019.

Our model well reproduces magnitude and vertical distribution of HCHO, isoprene and monoterpenes

abundance when wildfire is weak. Both measurements and model highlight the spatial homogeneity in

HCHO vertical profiles, suggesting a minor contribution of biogenic VOCs to HCHO VCD. With a

high sensitivity to near surface signals, MAX-DOAS measurements provides evaluation for modeled

biogenic HCHO variability. With a simple geometric approximation to retrieve HCHO from MAX-

DOAS measurements, we show that MAX-DOAS HCHO retrievals agree well with model results on

the seasonal trend of HCHO signal in both Fairbanks and Toolik Field Station. We also find good

correlation between MAX-DOAS HCHO and modeled isoprene emissions. Future work is warranted to

investigate MAX-DOAS retrievals with optimal estimation method and its comparison with model

results.

We further compared the model results to TROPOMI HCHO L2 product, reprocessed with background

HCHO VCD and AMF using GEOS-Chem model output. GEOS-Chem provides HCHO vertical

profiles a priori and background columns in a higher horizontal and vertical resolution than TM5-MP

CTM, the default model used in TROPOMI HCHO product. GEOS-Chem profiles includes the wildfire emission of the corresponding year, which TM5-MP did not, could be another advantage of GEOS-Chem. These advantages of GEOS-Chem model may improve the reliability of reprocessed TROPOMI

HCHO column ($VCD_{SAT,GC}$). We find that TROPOMI HCHO $VCD_{SAT,GC}$ in a mild wildfire summer is dominated by background HCHO $VCD_{0,GC}$ from methane oxidation. We find that wildfires have a larger contribution to HCHO total column than biogenic emissions, even in a year with mild wildfires. This result is in part due to the direct emission of HCHO from wildfires, and in part due to the slow and small production of HCHO from isoprene and monoterpenes oxidation under low $NO_x$ conditions. We

find that HCHO VCD from biogenic VOC is too small for TROPOMI to be able to detect.

For the year with large wildfires (2019), we find that TROPOMI and model show good agreement on magnitude and spatial pattern of HCHO VCD, and wildfire becomes the largest contributor to HCHO VCD inside fire-related enhancements. To a large extent this is driven by the direct emission of HCHO

from wildfires. We consider this good agreement to be partly fortuitous, due to uncertainties associated with satellite retrieval in smoke conditions, emissions strength and speciation, and detailed chemical mechanism for HCHO production. However, we show that wildfire emission signals are detectable in TROPOMI HCHO product, making TROPOMI a semi-quantitative tool to constrain wildfire emissions in Alaska. As the Arctic and boreal region continue to warm, we expect HCHO VCD in Alaska will

continue to be driven by wildfires and background methane oxidation.



*Acknowledgements.* T.Z., J. M. and W.R.S. acknowledge funding from NASA 80NSSC19M0154. We thank Dylan Millet, Xiaoyi Zhao for helpful discussions.

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
