# Peer review of "Source and variability of formaldehyde (HCHO) at northern high latitude: an integrated satellite, ground/aircraft, and model study"

_Atmospheric Chemistry and Physics, 2021_

## Author Comment (AC1)

Response to Reviewer #1

We are grateful to the reviewer for the valuable comments that facilitate the important improvements of the original manuscript. We list the point-by-point responses below. The reviewer's comments are marked black and our responses are marked dark blue. Line numbers refer to the discussion paper acp-2021-820. We attach the updated figures and supplementary information in the end.

Overall, this is an interesting and well written paper addressing VOC production and measurement in an understudied region. I think it is suitable for publication in ACP after some minor revisions. I think the primary areas that need to be addressed are:

The MAX-DOAS section. The MAX-DOAS geometric approximation for VCDs requires that the bulk of the trace gas column be above the scattering height. At line 373 you note that the HCHO column has a large fraction above the lowest kilometers, which already brings the validity of the geometric approximation into question. Moreover, you note that that you're looking at presenting optimally estimated profiles in a follow-up paper. In my view, the vertical profiling capability of the MAX-DOAS is its chief advantage as a ground-based measurement technique. It could provide you with some useful information to compare in this paper to ATOM results, and then to modelled profiles. I really don't think that geometric approximated MAX-DOAS VCDs (even if valid) are adding much to your discussion. It would be better either to leave the MAX-DOAS results out and save them for your follow-up paper, or (best case scenario!) incorporate the full optimally estimated profiles into this paper to take advantage of the MAX-DOAS's full capability.

We agree with the reviewer and will save the MAX-DOAS part from our follow-up paper.

The uncertainty section (4.4). This section could be tidied up and incorporated into your other results sections. You list many examples of uncertainty in different parameters from different papers, and yet I am still a little unclear on how you arrive eventually at the 90 % and 35 % uncertainty values for fire-free and fire-influenced scenarios. It would be great to spell out exactly how you incorporate each uncertainty term to calculate the final uncertainty. I also think you should do this earlier in the results section. This would aid your discussion of agreement between TROPOMI and GEOS-Chem by allowing you to specify whether/when/where you find agreement between the two less/greater than the TROPOMI uncertainty. You could help this further by including in your map plots (Figs 4,5 and 6) difference maps (GEOS-Chem minus TROPOMI or vice versa) to visually see where the agreement is below the uncertainty and/or less than the TROPOMI detection limit.

All the uncertainty values are directly from TROPOMI L2 HCHO ATBD. We reorganized the uncertainty part by:
1. Add a paragraph at the end of section 2.1: "We estimate the total uncertainty of reprocessed TROPOMI HCHO vertical column to be ≥ 90% for fire free region (TROPOMI L2 HCHO Algorithm Theoretical Basis Document, https://sentinels.copernicus.eu/documents/247904/2476257/Sentinel-5P-ATBD-HCHO-TROPOMI.pdf/db71e36a-8507-46b5-a7cc-9d67e7c53f70?t=1646910030856, and

references therein). This includes 75% of uncertainties from the $AMF_{SAT}$, 25% from $dSCD_{SAT}$ and 40% from $VCD_{0,SAT}$. The uncertainties in regions with strong fire are estimated to be $\geq 35\%$, including 30% of uncertainties from $AMF_{SAT}$, 15% from $dSCD_{SAT}$ and 10% from $VCD_{0,SAT}$. The relative lower uncertainties reflect much stronger VCDs in these wildfire regions. "

2. Remove the uncertainty section 4.4 and add a paragraph at the end of section 4.3 to address the uncertainty associated with retrievals in wildfire region: "Satellite retrievals of HCHO in wildfire region remains as a major challenge. One source of uncertainty stems from a priori profiles used in AMF calculation (Kwon et al., 2017). We find that for regions with heavy smokes, our calculated GEOS-Chem $AMF_{GC}$ is 50% lower than the $AMF_{SAT}$ in the operational product, due to the difference in HCHO vertical profiles (Figure S3). As a result, our reprocessed HCHO VCD product, $VCD_{SAT,GC}$, is higher than the operational product by $3\text{-}5\times10^{15}$ molecules $cm^{-2}$ in heavy smoke regions in July of 2019 (Figure S10). Another uncertainty lies in the aerosol optical properties. Wildfire smoke is a major source of brown carbon (June et al., 2020). As current retrieval algorithm for HCHO does not account for absorbing aerosols, it can reduce the sensitivity of satellite measurements to atmospheric layers below and above the aerosol layer, leading to a smaller AMF by 20-30%. (Jung et al., 2019; Martin et al., 2003)."

Minor corrections:

- Abstract line 1: spell out formaldehyde for the first time in the abstract too

Fixed.

- Lines 31-33: remove "to" in front of all the percentage ranges

Fixed.

- Lines 37-38: Sentence starting with "The source…" is repetition of previous information

We remove this sentence to avoid confusion.

We revise the part in abstract as "For the year with low wildfire activity (e.g., 2018), we find that HCHO VCDs are largely dominated by background HCHO (66-71%), with minor contributions from wildfires (20-32%) and biogenic VOC emissions (8-10%). For the year with intense wildfires (e.g., 2019), summertime HCHO VCD is dominated by wildfire emissions (50-72%), with minor contributions from background (22-41%) and biogenic VOCs (6-10%). In particular, the model indicates a major contribution of wildfires from direct emissions of HCHO, instead of secondary production of HCHO from oxidation of larger VOCs. We find that the column contributed by biogenic VOC is often small and below the TROPOMI detection limit, in part due to the slow HCHO production from isoprene oxidation under low $NO_x$ conditions.".

- Line 44: "show" not "shows"

Fixed.

- Line 51: remove "a significant amount of", it is subjective without quantification.

We now revise the sentence as:
"Volatile organic compounds (VOCs) emitted from terrestrial vegetation play a major role in air quality and chemistry-climate interactions (Guenther et al., 1995)."

- Line 58: remove "After these biogenic… … atmosphere", filler and not necessary for the flow of the sentence

The sentence at line 56-59 is modified to be "Primary biogenic VOCs, including both isoprene (2-methyl-1,3-butadiene, $C_5H_8$) and monoterpenes (a class of terpenes that consist of two isoprene units, $C_{10}H_{16}$), rapidly producing HCHO through oxidation after emitted to the atmosphere (Millet et al., 2006; Palmer et al., 2006)."

- Line 61: LAI already defined

Fixed.

- Line 68: remove "been"

Fixed

- Line 77-79: Reword the sentence beginning with "This high…". It reads like you are saying, in the end, there's an important role of climate warming on climate, which is tautological.

The sentence is modified to be "The high temperature sensitivity suggests an important role of climate warming on BVOC emissions."

- Line 81: HCHO already defined

The sentence is changed to be "HCHO serves as an important indicator of BVOC emissions on regional and global scales (Millet et al., 2006)."

- Line 88-89: Reword. It reads like "in regions where BVOC emissions are dominated by… the variation of BVOC emissions", again, tautological.

The sentence is modified to be "A number of studies use satellite-based observations of the HCHO column density to quantify regional and global isoprene emissions in vegetated regions (Guenther et al., 2006; Millet et al., 2008; Palmer et al., 2003, 2006; Stavrakou et al., 2009, 2014), and their interannual variability (De Smedt et al., 2010, 2015; Stavrakou et al., 2018, 2015, 2014; Zhu et al., 2017; Bauwens et al., 2016)"

- Line 90: Not clear how this sentence connects to previous paragraph. For example of what?

We remove this sentence to avoid confusion.

Instead, we cite Bauwens et al.(2016) and Stavrakou et al.(2018) in the previous sentence as "A number of studies use satellite-based observations of the HCHO column density to quantify regional and global isoprene emissions in vegetated regions (Guenther et al., 2006; Millet et al., 2008; Palmer et al., 2003, 2006; Stavrakou et al., 2009, 2014), and their interannual variability (De Smedt et al., 2010, 2015; Stavrakou et al., 2018, 2015, 2014; Zhu et al., 2017; Bauwens et al., 2016)".

- Line 128: remove "First", unnecessary

Fixed.

- Line 131: Not sure about "accuracy". (A) accuracy is hard to verify, as opposed to precision, and (B) I think the more important point is that ground based measurements are closer to being in-situ with, and therefore more sensitive to, the trace gas source.

The ground-based MAX-DOAS part is now removed in the revised text.

- Line 134-135: MAX-DOAS measurements are also really hard to interpret in cloudy and high AOD conditions. You say so yourself later when you omit MAX-DOAS measurements from the most smoke effected periods.

The ground-based MAX-DOAS part is now removed in the revised text.

- Line 167: "transfer" not "transport"

Fixed.
- Line 213: remove "that"

Fixed.

- Line 222: In the methods section, I would state that the reprocessed VCD has differences to TROPOMI VCD, rather than "advantages". Stating "advantages" starts to confuse results with methodologies.

Fixed.
- Line 226: Again, save this information about how your method leads to an improvement, for the results.

We removed and reorganized the paragraph (line 222-228) to the conclusion section 5: "We further compared GEOS-Chem results with TROPOMI HCHO L2 product, reprocessed with background HCHO VCD and AMF using GEOS-Chem model output. The reprocessed product may benefit from the finer horizontal and vertical resolution of GEOS-Chem than TM5-MP model, as well as the year-specific wildfire emissions……"

- Line 238: why different averaging times?

HCHO sampling is applied a 1-minute average. Isoprene and monoterpenes are sampled in 3-5 minutes interval and interpolated to 1-minute average.

We reorganized the ATom data introduction section 2.2. The averaging time part is now: "We make use of 1-minute averaged measurements of HCHO, isoprene, monoterpenes ($\alpha$-pinene and $\beta$-pinene) and the sum of methyl vinyl ketone and methacrolein (MVK+MACR). HCHO measurements sampled in 1-Hz frequency were made by laser induced fluorescence by the NASA In Situ Airborne Formaldehyde (ISAF) instrument (Cazorla et al., 2015). Isoprene and monoterpenes were measured by two instruments: one by the University of Irvine Whole Air Sampler WAS) followed by laboratory Gas Chromatography (GC) analysis, sampled every 3-5 minutes (Simpson et al., 2020); another by the National Center for Atmospheric Research (NCAR) Trace Organic Gas Analyzer (TOGA), sampled every 2 minutes with a 35-seconds integrated sampling time (Apel et al., 2021). MVK and MACR were measured by TOGA. These measurements are interpolated to 1-minute time resolution for model comparison. "

- Line 252-253: Why average to 2 hours for a 3-hour window? Why not just average all results from 12:00 to 15:00 (if you end up keeping the MAX-DOAS results in)?

The ground-based MAX-DOAS part is removed.

- Line 257: state why you would want to choose the highest elevation.

The ground-based MAX-DOAS part is removed.

- Line 264: I think shift this first sentence to be the second, the second sentence of the paragraph introduces the section better.

We replace the position of the first and the second sentence. Also, we upgrade our model to be GEOS-Chem v12.7.2 to avoid a bug in v12.5.0 which makes the simulation fail to read boundary conditions.

Now the first two sentences are "GEOS-Chem is a 3-D global chemical transport model driven by Modern-Era Retrospective analysis for Research and Applications, Version 2 (MERRA-2) by the Global Modeling and Assimilation Office (GMAO) at NASA's Goddard Space Flight Center (Rienecker et al., 2011), at a horizontal resolution of 0.5° × 0.625° and 72 vertical layers from surface to 0.01 hPa. Here we use GEOS-Chem v12.7.2 (http://wiki.seas.harvard.edu/geos-chem/index.php/GEOS-Chem_12#12.7.2 ), with an update on cloud chemistry (https://github.com/geoschem/geos-chem/issues/906). "

- Line 279: "have" not "has"

Fixed.

- Line 282: "BVOC emissions are calculated using", not "follows" – follows sounds jargonistic

Fixed.

- Line 302: "has" not "have"

Fixed.

- Line 302: Might be worth noting here whether, despite extensive validation, any extensive validation exists in this kind of environment.

This sentence has been modified to "This version of isoprene chemistry in GEOS-Chem has been extensively evaluated by recent field campaigns and satellite observations over southeast US (Fisher et al., 2016; Travis et al., 2016), including HCHO production from isoprene oxidation (Zhu et al., 2016, 2020; Kaiser et al. 2018). To our knowledge, this chemistry has not been evaluated at northern high latitude."

- Line 308-309: save for results

We remove the sentence at line 308-309 and add a sentence in the section 4.2: ". The widespread biogenic HCHO enhancement can be in part explained by the slow photooxidation in Alaska and low HCHO yield under low $NO_x$ conditions (~25-35 pptv near surface in GEOS-Chem) (Marais et al., 2012)."

- Line 324: Guide the reader with approximate altitude ranges in the text

We change the sentence to be "We show that the measured HCHO mixing ratio decreases exponentially from <2 km near surface (405 pptv) to the ~10 km upper troposphere (100 pptv)."

- Line 339: "reproduces", not "well reproduce"

Fixed.

- Line 342: "mixing ratios are" not "mixing ratio is"
- Line 343-344: Not clear, do you mean in the lowest 2 km?

Fixed.

- Line 346: First sentence is unnecessary and emotive, rendering the second sentence repetitive.

We think that the first sentence delivers the key information of this paragraph, so it might help reader understand the paragraph easier.

- Line 352: This suggests a minor contribution in most of Alaska, but perhaps not everywhere?

We change the sentence to be "Such spatial discrepancies between HCHO and isoprene/monoterpenes suggest a minor contribution of biogenic VOC emissions to HCHO column density over Alaska during summertime."

- Line 414: Give an example to show how "high" is "high", perhaps by comparing to other parts of the world, to some threshold, by relationship to uncertainty or the detection limit.

We now remove this sentence to avoid confusion.

We change the sentence to be "Over Alaska domain, HCHO VCD$_{SAT,GC}$ peaks around the interior Alaska boreal forest region (Figure S1), with VCD$_{SAT,GC}$ as $3.7 \times 10^{15}$ molecules cm$^{-2}$ in July; near north slope and Gulf of Alaska, VCD$_{SAT,GC}$ is around $2 \times 10^{15}$ molecules cm$^{-2}$ in July."

- Line 419: "of" not "for"

Fixed.

- Line 420: reword to "May to August 2018"

Fixed.

- Line 423: remove "largely", it is unnecessary

Fixed.

- Line 424-425: You say "stems from", but all you've proven is that the HCHO predominantly "resides in" the lowest atmospheric layers. In fact, you highlight the large contributions of background methane oxidation which may not necessarily stem from the lowest layer at all – methane could be transported from long-range including higher atmospheric layers.

We change the sentence to be "As the majority of HCHO VCD **resides in** lowest atmospheric layers (Figure 1),……"

- Line 425: can you comment on the extent to which fewer plants (presumably lower BVOCs) and more long-lasting snow (higher albedo, more retrieval problems) could contribute to the lower HCHO VCD in elevated regions?

The lower HCHO VCD in elevated region is mainly due to the lower HCHO background column, which is a model result. The dVCD over elevated region is in the similar magnitude as over the northern Pacific.

We add a sentence in section 4.1: "The spatial pattern of $VCD_{0,GC}$, most noticeable in July, is driven by the geography in Alaska, instead of surface vegetation or snow."

- Line 429: What causes this enhanced methane oxidation?

We revise the sentence (line 428-431) in section 4.1 to be "Enhanced methane oxidation likely results from the increase of water vapor and therefore OH production, leading to a higher HCHO production via $CH_3O_2$ + NO reactions near surface and $CH_3O_2$ + $CH_3O_2$ at higher altitudes."

- Line 443: remind the reader what a negative dVCD physically represents.

We add a sentence behind the sentence in line 443: "Negative values reflect the fact that averaged HCHO $dVCD_{SAT}$ is close to zero as a result of reference sector correction (TROPOMI L2 HCHO ATBD)."

- Line 450: I'm unclear on the relationship of ideas in this paragraph. How does "widespread HCHO enhancement" follow from the first sentence, then on to saying that HCHO production is actually suppressed by low NOx levels? In addition, please clarify quantitatively what you mean by low and high NOx

We clarify the relationship between widespread HCHO enhancement and low NOx level in section 4.2 : "The widespread biogenic HCHO enhancement can be in part explained by the slow photooxidation in Alaska and low HCHO yield under low $NO_x$ conditions (~25-35 pptv near surface in GEOS-Chem) (Marais et al., 2012). Indeed, the HCHO production from isoprene and monoterpene emissions is lower under low $NO_x$ conditions than high $NO_x$ conditions (~ 1 ppbv) by a factor of 10 after 24 hours of oxidation, and it only reaches 20% of its 5-day cumulative yield, leading to a suppressed but prolonged HCHO production (Marais et al., 2012)."

- Line 472: This small section is mostly repetition of ideas in the previous paragraph, it can be incorporated or removed

We merged it to the previous paragraph in section 4.2: "...... As a result, $dVCD_{GC,Fire}$ contributes to 20-32% of $dVCD_{GC}$, while $dVCD_{GC,Bio}$ contributes to 8–10% of $dVCD_{GC}$. Wildfire and biogenic emission are both important for $dVCD_{GC}$ and most active in central boreal forest region, posing a challenge to attribute TROPOMI $dVCD_{SAT,GC}$ to individual sources."

- Line 489: First short sentence not needed. Also, reword the next sentence to have "in the 2019 Alaskan summer."

The second sentence already contains "2019 Alaska summer". We remove the first sentence and change the second sentence to be "Figure 4(a) shows monthly $VCD_{SAT,GC}$ in the 2019 Alaskan summer."

- Line 492: add "the" between than and TROPOMI

Fixed.

- Line 497: "sources" not "source". Also, be quantitative instead of simply saying "Much lower…"

1. Fixed
2. We change the sentence (line 497-498) to be: "In contrast, HCHO VCD outside of the central Alaska are close to the background level, with little enhancement on background HCHO"

- Line 505: Be quantitative instead of simply saying "We find little change"

We change the sentence to be: "We find little change on $VCD_{0,GC}$ and $dVCD_{GC,Bio}$ (~$2 \times 10^{14}$ molecules cm$^{-2}$) from 2018 to 2019 summer in model sensitivity tests,……"

- Line 511-512: Why are the biogenic emissions higher by a factor of "1-2"? You have the numbers there, surely it is larger by a factor of 498/374 exactly?

We recalculate the values based on the new simulation.

We change the sentence to be: "Consequently, $dVCD_{GC,Fire}$ is higher than $dVCD_{GC,Bio}$ by a factor of 10 in 2019 Alaska summer, despite that NMVOC from wildfires (498 GgC) are only higher than biogenic emissions (389 GgC) by 30%."

- Line 534: Don't have "etc", be specific.

Fixed.

- Conclusions: I think you want to start your conclusions with positive results, what you want people to take away from this paper, not another summary of the previous literature. Imagine you get to the end of the paper, and after all that reading the first thing you see in the conclusion is "VOC emissions… remain poorly quantified…". No – tell the reader why the work you've done is great! Tell them how you've helped close a literature gap, don't highlight how one is still open. To achieve that, you can significantly shorten your conclusion, cutting it to the most salient points only.

We now revised the conclusion as:

"The Arctic/boreal terrestrial ecosystem is undergoing rapid changes in recent decades, but VOC emissions from Arctic and boreal vegetation and wildfires remains poorly quantified, limiting our capability for understanding biosphere-atmosphere exchange in this region and its feedback on Arctic climate and air quality. In this work, we use satellite-based observations of HCHO VCD from the TROPOMI instrument on-board S5P satellite, combined with a nested grid

chemical transport model, to examine the source and variability of HCHO VCD in Alaska for the summers with low fire activities (2018) and high fire activities (2019).

We first evaluate the GEOS-Chem nested simulation ($0.5° \times 0.625°$) with *in-situ* airborne measurements in Alaska from the ATom-1 mission. We show reasonable agreement between observed and modeled HCHO, isoprene, monoterpenes and the sum of MVK+MACR in the continental boundary layer. In particular, HCHO profiles show spatial homogeneity in Alaska, suggesting a minor contribution of biogenic emissions to HCHO VCD.

We further compared GEOS-Chem results with TROPOMI HCHO L2 product, reprocessed with background HCHO VCD and AMF using GEOS-Chem model output. The reprocessed product may benefit from the finer horizontal and vertical resolution of GEOS-Chem than TM5-MP model, as well as the year-specific wildfire emissions. We find that reprocessed TROPOMI HCHO $VCD_{SAT,GC}$ is dominated by background HCHO $VCD_{0,GC}$ from methane oxidation in a mild wildfire summer. Wildfires have a larger contribution to HCHO total column than biogenic emissions, even in a year with mild wildfires. This result is in part due to the direct emission of HCHO from wildfires, and in part due to the slow and small production of HCHO from isoprene and monoterpenes oxidation under low $NO_x$ conditions.

For the year with large wildfires in Alaska (2019), we find that TROPOMI and model show good agreement on magnitude and spatial pattern of HCHO VCD, and wildfire becomes the largest contributor to HCHO VCD. Model sensitivity suggests the direct emission of HCHO from wildfires accounts for the majority of HCHO VCD. While the emission factor of HCHO from wildfires (1.86 g/kg dry matter for boreal forest) applied in our model largely agree with field measurements, the role of secondary production of HCHO is likely underestimated due to unaccounted VOCs and underrepresented plume chemistry. We show that wildfire signals can be detected by TROPOMI HCHO product, making TROPOMI a semi-quantitative tool to constrain wildfire emissions in Alaska given the large uncertainties associated with HCHO retrieval in wildfire plumes. As the Arctic and boreal region continue to warm, we expect HCHO VCD in Alaska continues to be driven by wildfires and background methane oxidation.

Quantifying HCHO at northern high latitude can be further improved in several aspects. First, we show that background signal, often taken from model output, can be dominant in final product of HCHO VCD. However, model results differ significantly on HCHO even over Pacific Ocean (Figure S8), leading to a large uncertainty in the final satellite product in this region. Second, reference sector correction represents another major uncertainty (Zhu et al., 2020). This is particularly a problem for Alaska, as it lies in the reference sector defined by most retrieval algorithms (González Abad et al., 2015; De Smedt et al., 2018).  Any systematic bias in Alaska can propagate to retrievals in other regions. Third, pristine regions can also be influenced by wildfire plumes, which can largely impact HCHO retrieval. Future work is warranted to improve HCHO retrieval and therefore our understanding of HCHO at northern high latitude."

---

## Author Comment (AC2)

Response to Reviewer #2

We are grateful to the reviewer for the valuable comments that facilitate the important improvements of the original manuscript. We list the point-by-point responses below. The reviewer's comments are marked black and our responses are marked dark blue. Line numbers refer to the discussion paper acp-2021-820. We attach the updated figures and supplementary information in the end.

This paper presents model evaluations of HCHO against a combination of satellite, ground, and aircraft observations in a very sensitive area but rarely studied in terms of atmospheric composition. It is in general well written and fits well within the scope of ACP. It will add important insights regarding HCHO source and variability at high latitudes. I'd recommend it for publication. A few concerns and comments are listed below for considerations for clarification or potential improvement:

1. Isoprene (and monoterpenes) may not be good tracers to directly evaluate their emissions given its short lifetime. Do MVK+MACR observations available during Atom? They would provide a more regionally representative signal for isoprene emissions.

We now include MVK+MACR measurements from ATom-1 aircraft campaign for our model evaluation. Figure 1 is updated as follows.

[Figure]

We add a sentence in model-ATom comparison section 3: "In addition, modeled MVK+MACR shows average mixing ratio of 78 pptv while observations show 38 pptv, providing additional constraints on isoprene oxidation."

2. For model evaluation with Atom-1: why was 1 hour averaged model output used? The model was running using 10 min /20 min time steps? Would a higher time resolution comparison better help resolve the vertical profiles?

We now change the model output to 1-min time resolution using Planeflight diagnostics. We revised the text as:
"For the comparisons between observations and model shown below, we sample the model output along the flight track at the flight time with 1-min time resolution."

3. Figure 1: The model shows a somewhat large underestimate of HCHO in the free troposphere and the boundary layer? It seems so too for isoprene and monoterpenes? It would be worth emphasizing as they reflect some knowledge gaps that might be the first time shown in the literature.

Based on ATom and the model PlaneFlight diagnosis (mentioned in comment 2), we discussed the model underestimation above boundary layer (>2km). The HCHO VCD derived from model and ATom are comparable. We reorganized two paragraphs in ATom-model comparison section 3:

"Our model shows reasonable agreement with measurements in the boundary layer (<2 km). Modeled HCHO has a mean mixing ratio of 431 pptv, slightly higher than the observed value (405 pptv). Modeled isoprene has a mean mixing ratio of 227 pptv in the boundary layer, in agreement with observed values given the large variability of observations. Both observations and model show significantly less monoterpenes compared to isoprene, on the order of tens of pptv. In addition, modeled MVK+MACR shows average mixing ratio of 78 pptv while observations show 38 pptv, providing additional constraints on isoprene oxidation.

Our model tends to underestimate HCHO above boundary layer (>2 km). We show in Figure 1 that mean modeled HCHO is 98 pptv at 3-6 km, and ~46 pptv at 6-10 km, compared to observed values of 212 pptv and 104 pptv respectively. The reason is unknown, but could be related to the large underestimate of $CH_3OH$ in the same region (Bates et al., 2021). As a result, the model-derived HCHO VCD is likely lower than that calculated from ATom measurements, by $2.5 \times 10^{15}$ molecules $cm^{-2}$. Such bias may lead to a systematic bias on our estimate of background HCHO $VCD_0$ in this region."

4. Figure 2: why were the model results for the regional average used? Any justification that it should be this way? Or is it better than using the model output for the grid cell containing the station? Given the high-resolution model, I don't quite understand why such a regional average is needed.

The ground-based MAX-DOAS part is now removed in the revised text.

5. a) Figure 2 shows some interesting features of enhancements captured by the model. MAX-DOAS however seems quite a noise although it is hourly data. Can the comparison be done more quantitatively while still being able to factor in MAX-DOAS instrument uncertainty? How does the model perform in non-fire conditions vs fire influence conditions? Can any quantitative results be interpreted here? Would the MAX-DOAS be useful to compare to the TROPOMI HCHO products directly?

The ground-based MAX-DOAS part is now removed in the revised text.

b) Line 385: if the detection limit of MAX-DOAS is 1e15 molecules cm-2, then there'd be only a few data points above the detection limit in Figure 2? Am I interpreting it correctly?

The ground-based MAX-DOAS part is now removed in the revised text.

      c) Figure 2: some panels lack y scale.

The ground-based MAX-DOAS part is now removed in the revised text.

6. Fire influence in the model: Is that the fire influence in the nested domain, or is that global? Fire smoke in other regions may transport to AK and affect 2019 summer? Depending on how this sensitivity was set up (does it reflect the fire influence within the AK domain, or globally), it may be the reason why the fire VOC emission within AK is only a factor 1 to 2 higher than biogenic emissions, but $dVCD_{GC, Fire}$ is 10 X higher than $dVCD_{GC, Bio}$? i.e., Lines 510-513

The nested simulations are based on boundary conditions from a global run with wildfire and biogenic emission turned on. We made another nested simulation using a new boundary condition from a global run with wildfire and biogenic emission turned **off**.

We add a sentence at section 4.3: "Due to model sensitivity tests, intercontinental transport of wildfire emissions contributes a minor part of $dVCD_{GC,Fire}$ (~1% in interior Alaska, ~10% in southwest Alaska for 2019 July)."

7. From the comparison of TROPOMI HCHO VCD with the GEOS-Chem HCHO VCD, it seems the model is predicting HCHO well and there is no significant knowledge gap regarding HCHO from biogenic VOCs or fire smoke in Alaska? But from the comparison with Atom observations, the model seems underpredicting HCHO, while the MAX-DAOS comparison may not be too quantitative? How would these be reconciled, particularly regarding Atom and TROPOMI evaluations? Overall, I was hoping to see those evaluations could be done more quantitatively. How exactly does the model HCHO compare to observations? Does the model underpredict HCHO at the surface or throughout the troposphere, which seems to be the case when compared to Atom?

First, currently we cannot take advantage of the vertical profile optimal estimation of MAX-DOAS, so we have to remove this part.

Second, GEOS-Chem does have discrepancies on VOC vertical profiles comparing to ATom measurements, depends on VOC species. But for HCHO, only 35% of the difference is in <2km layer where dVCD mainly loads. Model sensitivity test shows that >90% of the HCHO total column discrepancy is on background.

We quantitatively discussed the underestimation of model HCHO in above 2 km layers, and the reasonable agreement with ATom in <2km layer. Please see the two paragraphs in our response for the comment 3.

At the last paragraph of satellite-model comparison section 4.3, we discussed the uncertainty that difference sources can introduce to TROPOMI VCD, including the reprocessing based on our model: "Satellite retrievals of HCHO in wildfire region remains as a major challenge. One source of uncertainty stems from *a priori* profiles used in AMF calculation (Kwon et al., 2017). We find that for regions with heavy smoke, our calculated GEOS-Chem $AMF_{GC}$ is 50% lower than the $AMF_{SAT}$ in the operational product, due to the difference in HCHO vertical profiles (Figure S3). As a result, our reprocessed HCHO VCD product, $VCD_{SAT,GC}$, is higher than the operational product by $3\text{-}5\times10^{15}$ molecules $cm^{-2}$ in heavy smoke regions in July 2019 (Figure S10). Another uncertainty lies in the aerosol optical properties. Wildfire smoke is a major source of brown carbon (June et al., 2020). As the current retrieval algorithm for HCHO does not account for absorbing aerosols, smoke can reduce the sensitivity of satellite measurements to atmospheric layers below and above the aerosol layer, leading to a smaller AMF by 20-30% (Jung et al., 2019; Martin et al., 2003)."

We discuss the challenges of comparing model with satellite HCHO in conclusion section 5: "Quantifying HCHO at northern high latitudes can be further improved in several aspects. First, we show that background signal, often taken from model output, can be dominant in final product of HCHO VCD. However, model results differ significantly on HCHO even over the Pacific Ocean (Figure S8), leading to a large uncertainty in the final satellite product in this region. Second, reference sector correction represents another major uncertainty (Zhu et al., 2020). This is particularly a problem for Alaska, as it lies in the reference sector defined by most retrieval algorithms (González Abad et al., 2015; De Smedt et al., 2018). Any systematic bias in Alaska can propagate to retrievals in other regions. Third, pristine regions can also be influenced by wildfire plumes, which can largely impact HCHO retrieval. Future work is warranted to improve HCHO retrieval and therefore our understanding of HCHO at northern high latitudes."

8. Lines 460-465: Here and a few other places claim the VCD is mostly driven by wildfire direct emission, rather than secondary production during fire conditions, but it is according to model sensitivity tests. The more quantitative comparison between model and observation may show the model is underpredicting HCHO vertical distribution (Item 7), and the satellite data comparison approach may be biased since it uses the model information for reprocessing (Item 9). I wonder if the observations and the model evaluations have any evidence to support that the direct fire emission of HCHO drives its VCD, rather than secondary productions.

Our discussion on the difference between model and ATom can refer to our response for comment 3.

Our discussion on the uncertainty in model-satellite comparison can refer to our response for comment 7.

We add recent in-situ measurements of HCHO emission factors in fire smokes, which is consistent to model fire HCHO emission factor, to introduction section 1: "The GFED4s inventory reports the HCHO emission factor to be 1.86g/kg dry matter for boreal forest fires and 2.09 g/kg dry matter for temperate forest fires, consistent with recent field measurements (Liu et al., 2017; Permar et al., 2021)"

We add a sentence in the conclusion section 5: "Model sensitivity suggests the direct emission of HCHO from wildfires accounts for the majority of HCHO VCD. While the emission factor of HCHO from wildfires (1.86 g/kg dry matter for boreal forest) applied in our model largely agree with field measurements, the role of secondary production of HCHO is likely underestimated due to unaccounted VOCs and underrepresented plume chemistry."

9.  a) I am a bit confused about the reprocessed TROPOMI HCHO VCD. My understanding is that it also uses information from GEOS-Chem (for a priori, background column, and AMF), and later the paper compares this reprocessed product with GEOS-Chem. Wouldn't that model information used to reprocess TROPOMI VCD cause some internal biases to the new data, so that the reprocessed product would be essentially similar to and dependent on the model? Can authors explain how it would or would not be the case, and would it affect the interpretation of HCHO VCD evaluation? In other words, is it a fair and independent comparison? The authors seem to agree with that by stating the TROPOMI products are a 'semi-quantitative tool' to constrain fire emission, which should be further clarified

The satellite retrieval indeed relies on model information to provide the final product. We here replace the model information in TROPOMI L2 product with a high-resolution model with year-specific wildfire emissions, for a more realistic representation of HCHO in the atmosphere.

The reprocess error from model includes two parts: background and AMF.

For background error, we discussed the model underestimation comparing to ATom, which is mainly in background. Our modification in the paper can refer to our response for comment 3.

For error from AMF and other sources, we discussed it at the last paragraph of satellite-model comparison section 4.3, which can refer to our response for comment 7.

b) Some common practices of evaluating satellite retrievals include smoothing the model with satellite averaging kernels so that they have the same vertical sensitivity or reprocessing the satellite data with a certain priori profile so that they reflect the measurements, rather than a priori information. It seems the model and satellite data in the work both use the same a priori and the AMF. Am I understanding it correctly? If so, how often the a priori is updated in the reprocessed product? Overall, it would be great if the method for reprocessed data can be further clarified, i.e., the exact difference between the reference sector correction of this study and the default.

Yes. We add a sentence at TROPOMI introduction section 2.1: "GEOS-Chem vertical profiles are updated hourly with collocated TROPOMI HCHO pixels."

The GEOS-Chem and S5P AMF are compared in Figure S7. The GEOS-Chem background versus S5P background is shown in Figure S8. Reprocessed TROPOMI HCHO VCD and S5P HCHO VCD are compared in Figure S10.

> c) The model seems to underpredict the HCHO vertical distribution relative to Atom field data, while the model is used to reprocess TROPOMI HCHO VCD. How does the HCHO underprediction relative to ATom affect the reprocessed VCD?

Our discussion on the model underestimation relative to ATom can refer to our response for comment 3.

The possible error sources of reprocessed VCD, including the model, can refer to our comment 7.

10. Line 100. The paper cites Liu et al. 2017 for HCHO EF. There are some new studies from recent aircraft campaign and they seem to support around 2 g/kg for HCHO EF (i.e., WE-CAN VOC emission paper Permar et al 2021 and recent FIREX-AQ AGU conference talks?). Would EF HCHO used in the model/GFED be consistent with those recent studies? It may be able to support that the fire emission in the model is simulated well.

We add this sentence to the introduction section 1: "Several studies have reported a similar level of HCHO emitted from wildfire plumes. Liu et al (2017) found formaldehyde as the second most abundant NMVOC from wildfires in western US, with an emission factor of 2.3 (±0.3) g/kg dry matter for temperate forests and a similar emission factor for boreal forest fires. WE-CAN aircraft measurement reports the HCHO emission factor in near-fire smoke plume to be 1.9 (±0.43) g/kg (Permar et al., 2021). "

The GFED HCHO emission factor is added in GEOS-Chem introduction section 2.3: "The GFED4s inventory reports the HCHO emission factor to be 1.86g/kg dry matter for boreal forest fires and 2.09 g/kg dry matter for temperate forest fires, consistent with recent field measurements (Liu et al., 2017; Permar et al., 2021)."

11. Lines 550 -555: I am not entirely sure how the uncertainties of reprocessed VCD were calculated by reading this part. Can the authors clarify it?

All the uncertainty values are directly from TROPOMI L2 HCHO ATBD.

We reorganized the uncertainty part by:
1. Add a paragraph at the end of section 2.1: "We estimate the total uncertainty of reprocessed TROPOMI HCHO vertical column to be ≥ 90% for fire free region (TROPOMI L2 HCHO Algorithm Theoretical Basis Document, https://sentinels.copernicus.eu/documents/247904/2476257/Sentinel-5P-ATBD-HCHO-TROPOMI.pdf/db71e36a-8507-46b5-a7cc-9d67e7c53f70?t=1646910030856, and references therein). This includes 75% of uncertainties from the $AMF_{SAT}$, 25% from $dSCD_{SAT}$ and 40% from $VCD_{0,SAT}$. The uncertainties in regions with strong fire are

estimated to be $\geq 35\%$, including 30% of uncertainties from $AMF_{SAT}$, 15% from $dSCD_{SAT}$ and 10% from $VCD_{0,SAT}$. The relative lower uncertainties reflects much stronger VCDs in these wildfire regions. "

2. Remove the uncertainty section 4.4 and add a paragraph at the end of section 4.3: "Satellite retrievals of HCHO in wildfire region remains as a major challenge. One source of uncertainty stems from *a priori* profiles used in AMF calculation (Kwon et al., 2017). We find that for regions with heavy smoke, our calculated GEOS-Chem $AMF_{GC}$ is 50% lower than the $AMF_{SAT}$ in the operational product, due to the difference in HCHO vertical profiles (Figure S3). As a result, our reprocessed HCHO VCD product, $VCD_{SAT,GC}$, is higher than the operational product by $3-5 \times 10^{15}$ molecules $cm^{-2}$ in heavy smoke regions in July 2019 (Figure S10). Another uncertainty lies in the aerosol optical properties. Wildfire smoke is a major source of brown carbon (June et al., 2020). As the current retrieval algorithm for HCHO does not account for absorbing aerosols, smoke can reduce the sensitivity of satellite measurements to atmospheric layers below and above the aerosol layer, leading to a smaller AMF by 20-30% (Jung et al., 2019; Martin et al., 2003)."

12. Section 2.2: how many vertical profiles were used from ATom. How could AWAS do 3-5 minutes average for isoprene or monoterpenes. That's be lots of samples for AWAS? A bit of clarification would be good.

We reorganized the ATom data introduction section 2.2: "During ATom-1, two flights sampled eight vertical profiles over Alaska during 1-3 August 2016. We make use of 1-minute averaged measurements of HCHO, isoprene, monoterpenes ($\alpha$-pinene and $\beta$-pinene) and the sum of methyl vinyl ketone and methacrolein (MVK+MACR). HCHO measurements sampled in 1-Hz frequency were made by laser induced fluorescence by the NASA In Situ Airborne Formaldehyde (ISAF) instrument (Cazorla et al., 2015). Isoprene and monoterpenes were measured by two instruments: the University of Irvine Whole Air Sampler WAS) followed by laboratory Gas Chromatography (GC) analysis, sampled every 3-5 minutes (Simpson et al., 2020), and the National Center for Atmospheric Research (NCAR) Trace Organic Gas Analyzer (TOGA), sampled every 2 minutes with a 35-second integrated sampling time (Apel et al., 2021). MVK and MACR were also measured by TOGA. These measurements are interpolated to 1-minute time resolution for model comparison. Within our study domain, there are 341 1-minute averaged mixing ratio values for HCHO, 101 and 231 for isoprene and $\alpha$-pinene/$\beta$-pinene from WAS, 337 for isoprene, $\alpha$-pinene/$\beta$-pinene and MVK/MACR from TOGA. The reported measurement uncertainties are $\pm10\%$ for HCHO, $\pm10\%$ for WAS isoprene and monoterpenes, $\pm15\%$ for TOGA isoprene and $\pm30\%$ for TOGA monoterpenes, $\pm30\%$ for MVK and $\pm20\%$ for MACR."

13. Line 523: ATBD not defined until next page.

Fixed.